# Application of Fixed-Length Ultrasonic Interferometry to Determine the Kinetics of Light-/Heat-Induced Damage to Biological Membranes and Protein Complexes

**Denis V. Yanykin** [1,2,*], **Maxim E. Astashev** [1,3], **Andrey A. Khorobrykh** [2], **Mark O. Paskhin** [1], **Dmitriy A. Serov** [1,3] **and Sergey V. Gudkov** [1]

1   Prokhorov General Physics Institute of the Russian Academy of Sciences, 38 Vavilova St., 119991 Moscow, Russia
2   Institute of Basic Biological Problems, FRC PSCBR, Russian Academy of Sciences, 2 Institutskaya St., Pushchino, 142290 Moscow, Russia
3   Institute of Cell Biophysics, FRC PSCBR, Russian Academy of Sciences, 3 Institutskaya St., Pushchino, 142290 Moscow, Russia
*   Correspondence: ya-d-ozh@rambler.ru

**Abstract:** This manuscript describes the application of a fixed-length ultrasonic spectrometer to determine the kinetics of heat- and photo-induced damage to biological membranes and protein complexes and provides examples of the test measurements. We implemented a measurement scheme using the digital analysis of harmonic signals. To conduct the research, the fixed-length ultrasonic spectrometer was modernized: the speed was increased; lighting was supplied to the sample cells; the possibility of changing the gas atmosphere and mixing the sample was given. Using solutions containing natural concentrations of deuterium oxide, a high sensitivity of the spectrometer was shown. The spectrometer performed well in the measurement of phase state of dimyristoylphosphatidylcholine liposomes, both in the absence and in the presence of additions, which are capable of changing the lipid properties (sodium dodecyl sulfate, palmitic acid, and calcium ions). The heat- and photo-induced changes in the state of photosystem II core complexes were demonstrated using a fixed-length ultrasonic spectrometer. Transitions at 35.5 °C, 43.5 °C, 56.5 °C, and 66.7 °C were revealed. It is proposed that the transitions reflect the disassembly of the complexes and protein denaturation. Thus, the present study demonstrates that a fixed-length ultrasonic spectrometer can be applied to determine the kinetics of heat- and photo-induced damage to biological membranes and protein complexes.

**Keywords:** ultrasound; fixed-length ultrasonic spectrometer; biological membranes; protein complexes; phase states; phase transitions; photosystem II

## 1. Introduction

Biological membranes are unique heterogeneous three-dimensional structures, consisting of proteins embedded in a lipid matrix, the architecture and stability of which are strongly related to their functions. One of the most important characteristics of biological membranes is their phase state, including the state of protein molecules integrated into the membrane. It is known that lipids, the main structural components of biological membranes, may have three phase states: liquid (liquid crystal), ripple and solid (gel) [1]. The phase transition point depends on many factors: the structure and packing of lipid molecules, hydrostatic pressure, salt concentration, and so on. Phase transitions of lipid membranes are the basis of important biological processes: the fusion of cells, osmosis, the transport of substances, etc. [2–7]. Proteins can also exist in several thermodynamically stable states, for example the native state, denatured state, and compact state of the molten globule with a loss of unique side chain packaging. The transitions between these states in

proteins also resemble the phase ones: with a smooth change in conditions (temperature, pH, or ionic strength), there are drastic changes in the structure of the protein molecule, the cause of which is also a large amount of weak interactions that determine the tertiary structure [8–13].

Despite the fact that the structure of the lipid bilayer is quite stable, individual phospholipid molecules have some freedom of motion. The structure and stability of the lipid bilayer depend on the lipid composition and change with temperature. Lipids form a semi-solid phase (gel, liquid crystal state) in the bilayer at temperatures below physiological. In this state, all motions of individual lipid molecules are strictly limited, phospholipids are arranged in a strictly ordered manner, and the hydrophobic hydrocarbon tails of phospholipid molecules are completely extended parallel to each other. As a result, the surface area of the membrane is greatly reduced. At temperatures above physiological, the hydrocarbon chains of fatty acids are in constant motion, due to the rotation of the hydrocarbon bonds of long acyl chains. In this liquid-crystalline (or liquid-disordered) state, the inner region of the bilayer resembles a sea of lipids in constant motion. In the liquid state, structural transitions are possible due to thermal motion: the molecules are bent; their parallelism is broken in some places. As a result, the surface area of the membrane significantly increases. At intermediate temperatures, biomembrane components, including proteins, exist in a liquid-ordered state [14]: the thermal motion is reduced in the acyl chains of the lipid bilayer, but lateral movement still occurs. These differences in the state of the bilayer are easily observed in monolipid liposomes. For example, the solid-to-liquid phase transition of the synthetic phospholipid dipalmitoylphosphatidyl choline occurs over a very narrow temperature range, down to as little as one degree Celsius (43 °C). At the same time, biological membranes, which contain many lipids with a variety of acyl chains, do not show fast phase transitions with temperature changes.

The thermal phase transition is the most-studied phase transition. At the same time, a number of compounds can affect the packing and ordering of phospholipids in a lipid bilayer membrane: local anesthetics, short alcohols, and hydrocarbons make the membrane more liquid (including the decrease in the phase transition temperature); at the same time, calcium ions, long alcohols, and hydrocarbons structure the membrane (including the increase in the phase transition temperature) [15,16]. In this case, it is correct to say that a chemotropic phase transition occurs. One of the important membrane modifiers is cholesterol [17,18]. Cholesterol molecules are located between phospholipid molecules. They order the bilayer in the liquid state and disorder it in the solid state, thus reducing the differences between the liquid crystal and solid crystal structures.

Currently, the main approaches for studying the state and phase transitions are differential scanning calorimetry, NMR spectroscopy, Fourier transform-infrared spectroscopy, quartz crystal microbalance with dissipation monitoring, etc. [19–33]. However, most of these approaches require expensive equipment and expendable materials. It is a good alternative to assess the state and phase transitions of membranes and protein molecules using an ultrasonic spectrometer [34–38]. The method of ultrasonic differential spectrometry combines high sensitivity, accuracy, and a small volume of the test sample [39]. The principle of measurement is to study the resonant properties of a composite resonator that includes a sample [40,41]. The method allows investigating the properties of solutions, colloids, suspensions, and emulsions. Measurement of the speed of sound in a sample allows one to automatically measure the compressibility of samples. The measurement of the compressibility of suspensions and emulsions is an approach that allows estimating the amount of water in the hydration shell of hydrophobic particles or the change in the number and state of these particles. In addition, the method makes it possible to measure acoustic impedance, excess enthalpy, miscibility, and the compatibility of mixtures. Compressibility, the derivative of volume with respect to pressure, is a physical parameter that characterizes the potentials of intermolecular interactions in a condensed medium and the response of the molecular structure to pressure. These two factors determine the instantaneous (high-frequency) and relaxation parts of the compressibility, respectively. The value of the

compressibility determines the phase velocity of sound in the medium, and its relaxation part determines the value of sound absorption. Therefore, simultaneous measurements of the velocity and absorption of ultrasound, as well as the density of the medium make it possible to determine the compressibility of the medium and its relaxation part with very high accuracy. In addition, the compressibility of an aqueous solution is highly dependent on the hydration of the solute. Almost all molecular processes (including phase transition) in solution are accompanied by a change in hydration, which can be recorded and characterized by the measurement of compressibility [42]. On the one hand, compressibility is a sensitive characteristic of the state of aggregation of a substance. On the other hand, a protein molecule and a lipid membrane are systems with a large number of internal degrees of freedom, being delimited from the medium. Therefore, they can be considered as a phase of small size, and their properties can be described in "macroscopic" terms [43].

One of the most accurate methods is the assessment of the state and phase transitions of biomolecules by changing the compressibility and absorption of ultrasound using a setup containing a fixed-length ultrasonic. To do this, we previously used a phase-locked loop (PLL) [41], which was later replaced by an alternative measurement scheme based on the methods of digital analysis of harmonic signals [44–46]. In the present work, the installation was further modernized, which made it possible to increase the speed, as well as to perform studies to determine the kinetics of heat- and photo-induced damage to biological membranes and protein complexes. The approach proposed in this work can be the basis for obtaining such important characteristics of a phase transition as the type of melting and denaturation of proteins, interfacial tension, and the rate constant of the elementary act of the growth and decomposition of a new phase, providing information about the physical properties of lipid membranes and protein molecules and on intermolecular interactions in aqueous solutions. This method combines high sensitivity, accuracy ($10^{-4}$%), and a small volume (100 Ml–1000 μL) of the test sample [39].

## 2. Materials and Methods

The dependence of the relative velocity of ultrasound in water on temperature at various concentrations of deuterium oxide were analyzed using different ratios of light water ("Legkaja voda, Standart", Komponent-Reactiv, Moscow, Russia) (without deuterium oxide) and ordinary water. The ordinary water was obtained by distillation and subsequent deionization of tap water.

Dimyristoylphosphatidylcholine liposomes were prepared by fifteen times of extrusion through a 200 nm membrane at room temperature [45]. Prior to this, the dimyristoylphosphatidylcholine (Sigma Aldrich, Merck Group, Darmstadt, Germany) was hydrated for several hours, followed by five freezing/thawing cycles. This procedure led to the formation of unilamellar vesicles [45,47–49]. Ultrasonic measurements were performed in the medium containing 50 mM KCl (Sigma Aldrich, Merck Group, Darmstadt, Germany), 20 μM EGTA (Sigma Aldrich, Merck Group, Darmstadt, Germany), 10 mM Tris-HCl (pH 8.5) (Sigma Aldrich, Merck Group, Darmstadt, Germany), and 1% ethanol (Chimmed Group, Moscow, Russia) (added with palmitic acid (Sigma Aldrich, Merck Group, Darmstadt, Germany)). The concentration of lipids used in the measurements was 1 mg/mL.

The isolation of photosystem II (PSII) core complexes was performed according to [50] with some modification [51]. The PSII membrane preparations used for the isolation of PSII core complexes were obtained in accordance with [52]. Removal of the water-oxidizing complex (WOC) from PSII core complexes was performed by 1 h of incubation in the presence of 5 mM $NH_2OH$ (Sigma Aldrich, Merck Group, Darmstadt, Germany) and subsequent purification on a Q-Sepharose column (GE Healthcare Bio Sciences, Uppsala, Sweden) [51]. Photoinhibition of "native" PSII core complexes was achieved by continuous illumination (λ = 400–800 nm, 1500 μmol photon $s^{-1}$ $m^{-2}$) provided by LED (JH-5WBVG14G24-Y6C, Ledguhon, Guangzhou Juhong Optoelectronics Co., Ltd., Guangzhou, Guǎngdōng, China) in a medium containing 50 mM MES (pH 6.5) (PanReac AppliChem, ITW Reagents, Glenview, IL, USA), 35 mM NaCl (PanReac AppliChem, ITW Reagents,

Glenview, IL, USA), and 0.003% (w/v) n-dodecyl-β-D-maltoside (PanReac AppliChem, ITW Reagents, Glenview, IL, USA) at a chlorophyll concentration of 18 μg/mL. The photoinhibition procedure was performed in the ultrasonic spectrometer cell. The ultrasound study of heat- and photo-induced damage to photosystem II core complexes was performed in the presence of 50 mM MES (pH 6.5), 35 mM NaCl, and 0.003% (w/v) n-dodecyl-β-D-maltoside at a chlorophyll concentration of 18 μg/mL (for heat-induced processes) or 7 μg/mL (for light-induced processes).

The Chl concentration was determined in 80% acetone (Chimmed Group, Moscow, Russia) [53]. The manganese content in the PSII preparations was controlled using the atomic absorption spectrometer KVANT-2A (Cortec, Moscow, Russia).

During the ultrasonic measurement, the samples in both the reference and experimental cells were mixed with a magnetic stirrer. All measurements were repeated at least three times.

## 3. Results

### 3.1. Setup

Figure 1 shows the structure of the measuring scheme of a fixed-length digital computer ultrasonic spectrometer. In comparison with the previous version [44], in order to overcome the limitations of the method associated with the speed of measuring the phase-frequency characteristic, the measurement channel of the experimental cell was upgraded. The directions of modernization were as follows:

1.　Acceleration of the operation of the controller that tunes the DDS generator: an 8-bit ATMega family (Atmel, Microchip Technology Inc., Chandler, AZ, USA) with a clock frequency of 16 MHz was replaced with a 32-bit SAM3 controller (Atmel, Microchip Technology Inc., Chandler, AZ, USA), that is replacing the Arduino Uno controller with the Arduino DUE controller.
2.　Transition from a circuit with one oscillator and one oscilloscope serving two channels and a switch to a circuit with one 125 MHz master oscillator and two DDS generators.
3.　Replacement of the digital oscilloscope in the channel of the experimental cell with a high-speed recorder with 32 MB of RAM and a USB3.0 interface. The recorder is started by an external trigger. The trigger signal is generated by the Arduino DUE.
4.　The measurement procedure in the channel of the experimental cell was changed. The Arduino DUE controller independently forms a grid of 15 frequencies and the 16th interval with the generator turned off. The entire frequency grid is automatically fed to the input of the experimental cell without the participation of a personal computer. The measurement result (in the form of a continuous recording with a sampling frequency of 50 MHz and eight million samples long) is transferred to a computer and analyzed. Thus, the procedure for measuring the characteristics of one resonant peak takes 160 ms, rather than 4 s as in the previous version.

In addition, the measuring cells were additionally equipped with magnetic stirrers (2 in Figure 2), LEDs (JH-5WBVG14G24-Y6C Ledguhon, Guangzhou Juhong Optoelectronics Co., Ltd., Guangzhou, Guǎngdōng, China), and capillaries for supplying or evacuating gases mounted in the covers (5 in Figure 2).

The program used to implement the algorithm for signal analysis (using digital signal processing) and control of the experimental conditions was developed with C++. The program performs the following tasks in real-time: (i) obtains raw digitized sinusoidal signals at the input and output of each cell, (ii) sets the frequency of the DDS generator, (iii) calculates the phase frequency dependence and amplitude transmission coefficient (AFC) and determines the position of the maximum of the resonant peak in the frequency range, (iv) visualizes the data, and (v) scans the temperature in the cells and controls the thermostat.

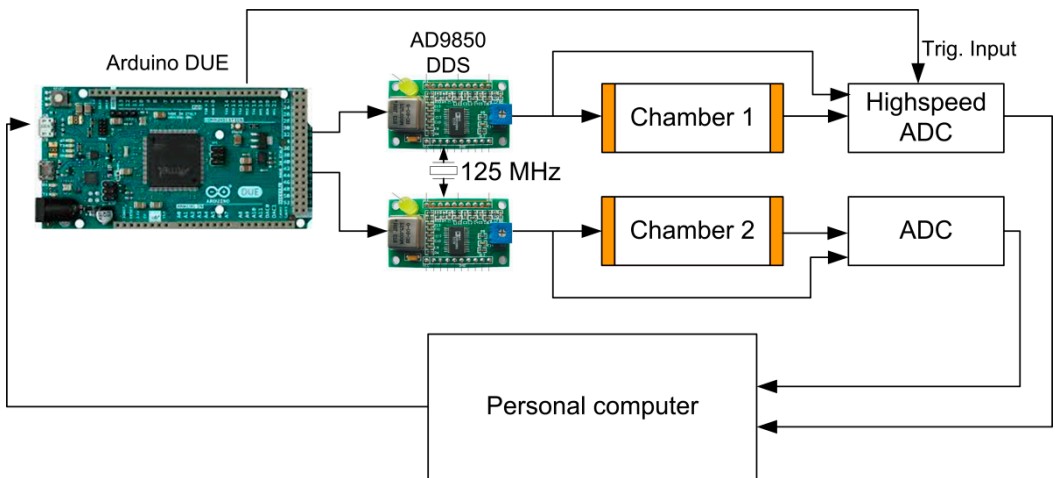

**Figure 1.** Block diagram of the setup containing a fixed-length ultrasonic spectrometer.

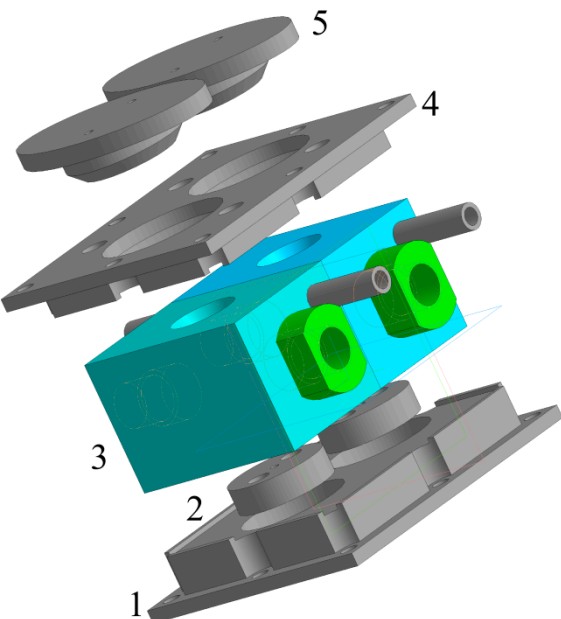

**Figure 2.** The 3D layout of the fixed-length ultrasonic spectrometer. (1) Bottom plate (PETG, 3D-printed); (2) magnet stirrer rotor (PETG, 3D-printed); (3) ultrasound measuring chambers (nickel-plated brass, machining); (4) top plate (PETG, 3D-printed); (5) caps (PETG, 3D-printed). DC brush motors, magnets 6 × 3, channels for supply/removal of gases to cells, and LEDs are not shown.

In comparison to the method described earlier [44], in the present work, the principle of determining the position of the resonance peak was changed (Figure 3). If the algorithm did not change much for the sample cell, then a simplified procedure was used for the reference cell. The phase frequency dependence near the resonant frequency was measured to perform steady-state measurement of the frequency and width of a resonant peak in a cell with a sample. The estimation of the position of the resonant peak was made by determining the position of the zero of the second derivative of the phase frequency dependence (Equation (1)). To obtain one point of the second derivative of the PFC, it is necessary to perform three dimensions of phase frequency dependence. This was repeated five times at small, regular frequency intervals. After that, the dependence of the second derivative of the phase frequency dependence was approximated by a straight line, which intersects the zero of the ordinate at the position of the resonant peak. In parallel, the first derivative was calculated (Equation (2)) for the calculation of the fork pitch at the

next determination of the position of the resonance peak (Equation (3)). The width of the resonance peak was calculated using Equation (4).

$$\phi''(f) = \frac{\phi(f + \Delta f_{[yellow]\phi}) + \phi(f - \Delta f_\phi) - 2\phi(f)}{\Delta f_\phi{}^2} \tag{1}$$

$$\phi'(f) = \frac{\phi(f + \Delta f_\phi) - \phi(f - \Delta f_\phi)}{2\Delta f_\phi} \tag{2}$$

$$\Delta f_\phi = a_\phi \frac{\pi}{4\phi'(f)} \tag{3}$$

$$\Delta f = \frac{\pi}{2\phi'(f)} \tag{4}$$

where $\Delta f_\phi$ is the fork pitch and $a_\phi$ is the trimming factor. We used $\Delta f$ to determine the $Q$-factor for the energy losses in our resonant system (Equation (5)):

$$Q = \frac{f}{\Delta f} \tag{5}$$

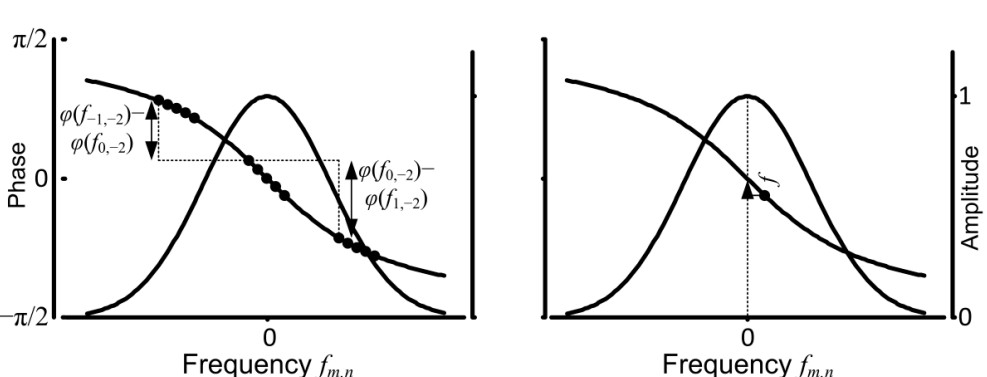

**Figure 3.** Comparison of the method for the determination of the position of the resonant peak in the cell with the test sample (**left**) and in the control cell with water (**right**).

As the $Q$-factor rises with losses falling, we used the $1/Q$ parameter to characterize the losses in the system due to scattering and viscosity.

As mentioned above, a simplified procedure was used for the reference cell, in which the phase was determined once for a frequency close to the frequency of the resonant peak. The current position of the resonant peak was determined by linear regression of the phase frequency dependence with a known slope. As a reference substance, it is better to use a known substance, for example pure water.

*3.2. Measurements*

Figure 4 presents the results of the measurement of the relative velocity of ultrasound in water. Line 1 indicates that the ultrasonic spectrometer had a good stability in all temperature ranges. Replacement of 50% (line 2) or 100% (line 3) light water in the experimental cell with ordinary water led to a decrease in the relative velocity of ultrasound. Moreover, this difference increased with increasing temperature. If we take into account the content of deuterium oxide in ordinary water (about 0.015%), the presented data confirm the high sensitivity of the ultrasound spectrometer.

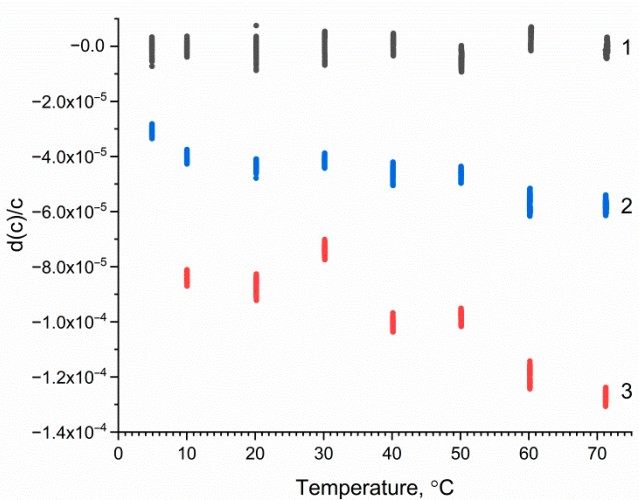

**Figure 4.** Dependence of the relative velocity of ultrasound in water on temperature at various concentrations of deuterium oxide. (1) Light water (without deuterium oxide), (2) 50% light water and 50% ordinary water (with a natural amount of deuterium oxide), and (3) ordinary water. There was light water in the reference cell. The ordinary water is deionized distilled tap water. c is the velocity of ultrasound in the reference cell; d(c) is the difference of the velocity of ultrasound in the sample cell and in the reference cell.

Figure 5 shows the influence of the addition of sodium dodecyl sulfate (SDS), palmitic acid, and $Ca^{2+}$ on the phase state of dimyristoylphosphatidylcholine (DMPC), a kind of phospholipid, which can be used to prepare liposomes. A sharp decrease of the ultrasound velocity at 23.85 °C reflects a change of the phase state of the liposomes. The addition of 20 μM SDS (a known agent, which destabilizes the membrane structure [54–61]) led to a slight decrease in the temperature of the phase transition (to 23.25 °C). Palmitic acid (20 μM), in contrast to the effect of SDS, increased the phase transition temperature and decreased the cooperativity of the transition (two phase transitions were observed: at 26.10 °C and 28.90 °C). These data are consistent with the previous data [62–66], including those obtained using ultrasonic measurements [67]. The addition of 200 mM $Ca^{2+}$ eliminates the SDS- and palmitic acid-induced changes, probably due to the binding of $Ca^{2+}$ to palmitic acid [46].

It is known that photosystem II is the pigment–protein complex, which consists of 20 protein subunits and about 20 lipid molecules [68–71]. The study of the state and phase transitions in such large complexes is a very difficult task, since the contribution of each component of the complex will overlap with the others. Figure 6A shows the kinetics of the ultrasonic velocity in the suspension containing untreated photosystem II (PSII) core complexes (black curve). The relative ultrasonic velocity gradually decreased with increasing temperature. The decrease in velocity with the temperature increase was nearly linear. However, in some parts of the kinetics, a decrease in the slope angle was observed, which may indicate changes in the protein complexes. The calculation of the derivative of the kinetics makes it possible to visualize the changes in the kinetics (Figure 6B). There are several regions of sharp growth on the derivative curve, which may indicate the decomposition of protein complexes into small components or a change in the structure of the peptides themselves. Several regions of this kind can be distinguished on the curve. The first starts at 43.5 °C, the second at 56.5 °C, and the third at 66.7 °C. Preliminary photoinhibition of the samples somewhat changed the character of the curves (Figure 6A,B, red curves). The relative ultrasonic velocity increased, which can indicate light-induced damage to the complexes (Figure 6A). The first shoulder on the derivative (observed at 43.5 °C) decreased greatly, which may indicate that those PSII components that undergo changes at this temperature were damaged by light prior to the ultrasonic measurement (Figure 6B). The curve shows a significant increase in the amplitude of the second shoulder (observed at 56.5 °C). Further, the character of both the changes in the

ultrasound velocity and their derivatives was practically the same in both types of samples (native and photoinhibited).

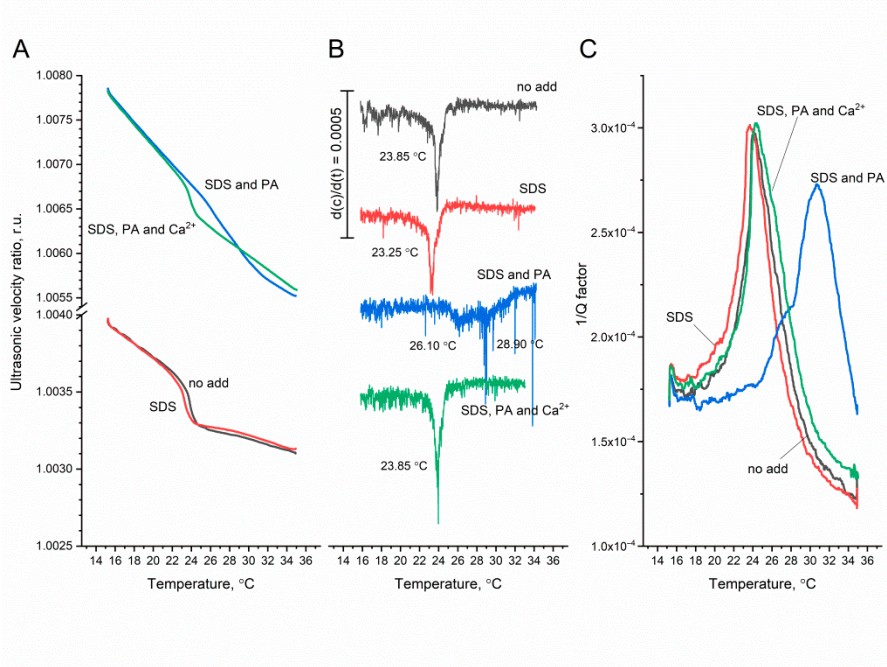

**Figure 5.** Effect of sodium dodecyl sulfate (SDS), palmitic acid (PA), and $Ca^{2+}$ on the phase state of dimyristoylphosphatidylcholine liposomes. Dependence of the relative velocity of ultrasound (**A**), its derivative (**B**), and the 1/Q-factor (**C**) on temperature. Measurements were performed in the medium containing 50 mM KCl, 20 μM EGTA, 10 mM Tris-HCl (pH 8.5), and 1% ethanol (added with palmitic acid).

Figure 6C describes the dependence of the Q-factor (which can describe the change in the shape or size of particles) on temperature. The figure shows that the initial value of the Q-factor for samples pre-treated with high-intensity light was higher than for untreated samples. This may indicate that PSII core complexes were damaged as a result of pre-illumination. When the temperature reached 33.5 °C, the Q-factor significantly increased in both types of samples and then did not change. At the same time, its value became the same for untreated and pre-illuminated preparations. It is important to note that the presented curves were the result of filtering the original curves (shown in the insert in Figure 6D). The original curves showed that the signal-to-noise ratio also changed with increasing temperature, which is reflected in Figure 6D. It is shown that the value of the Q-factor RMS began to increase with increasing temperature, which correlates well with the data presented in Figure 6A–C. The Q-factor RMS returned to the initial values when all the main processes, reflected in Figure 6B, ended.

It is known that photosystem II complexes depriving the water-oxidizing complex are very sensitive to photoinhibition [72–76]. Therefore, we decided to perform a study of the kinetics of photoinhibition using preparations with the WOC removed. In order to avoid additional thermal inactivation of the preparations, it was decided to conduct the experiments at a low temperature (≈6.4 °C). Figure 7 shows the photoinduced changes of the relative velocity of ultrasound in the suspension containing photosystem II core complexes with the removal of the WOC. Before illumination, the ultrasonic velocity remained stable.

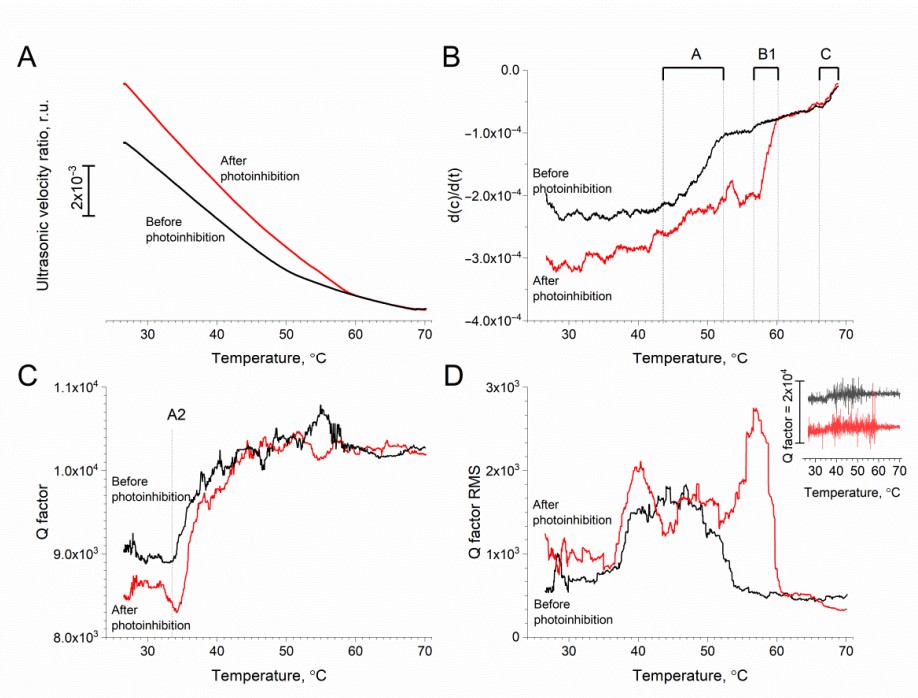

**Figure 6.** Ultrasound study of heat-induced damage to photosystem II core complexes before and after the photoinhibition procedure. (**A**) Relative velocity of ultrasound, (**B**) its derivative as a function of temperature, (**C**) the Q-factor, and (**D**) the Q-factor RMS as a function of temperature. Letters in (**B**,**C**) indicate transitions: A (43.5 °C), B1 (56.5 °C), C (66.7 °C) (the nomenclature accepted in [26] was used), and A2 (33.5 °C) (the nomenclature accepted in [28] was used).

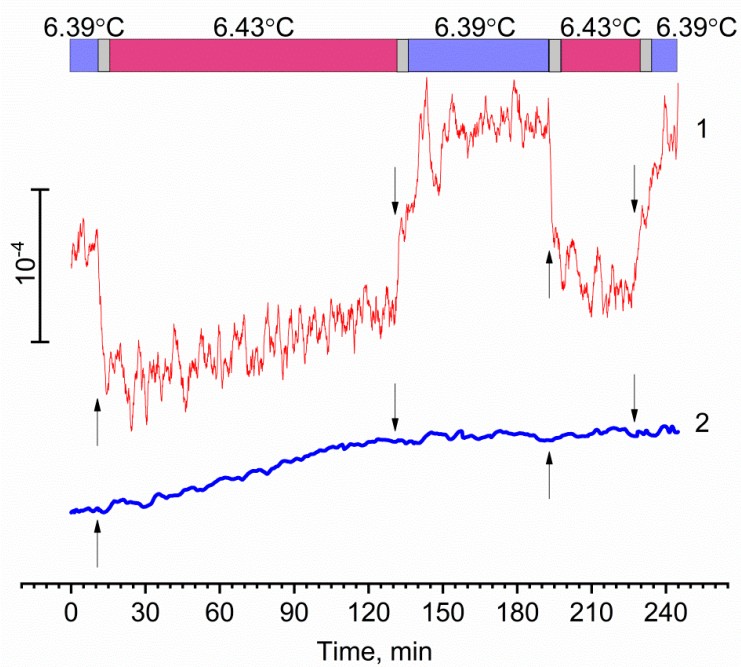

**Figure 7.** Photoinduced changes of the relative velocity of ultrasound in the suspension containing photosystem II core complexes with the removal of the water-oxidizing complex. The kinetics are presented before (1) and after (2) temperature compensation. The panel at the top of the figure shows the change of temperature in the cells induced by illumination. ↑ and ↓ represent the steady-state actinic light ($\lambda$ = 400–800 nm, 1500 µmol photon $s^{-1}$ $m^{-2}$) being on and off, respectively.

The kinetics of the photoinduced changes of the relative velocity of ultrasound can be divided into two parts: a fast phase, associated with sample heating, and a slow phase, reflecting photoinduced changes occurring in PSII complexes. The fast phase is a completely reversible decrease in the relative speed of ultrasound immediately after turning on the light, which is completely dependent on the heating of the sample ($\Delta t = 0.04$ °C). The slow phase is a gradual irreversible increase in the speed of sound as a result of the photoinhibition of the samples, which was not detected upon repeated illumination. Note that the rate of the light-induced increase of the ultrasound velocity after the second illumination was three-times lower that after the first one. These data correlate well with those presented in Figure 6A (increase in the speed of sound in the photoinhibited sample).

Thus, it was shown that the fixed-length ultrasonic spectrometer can be used to determine the kinetics of the light-/heat-induced damage to biological membranes and protein complexes.

## 4. Discussion

The development of technologies for the study of the phase state of biological samples concentrates on the increase of the resolution and on the decrease of the noise in comparison to the signal. Ultrasonic technologies are also constantly being developed [77–87]. However, the majority of the attention is paid either to fairly simple or model objects or to the assessment of the quality or readiness of food products, as well as research for medical purposes [34–40,45,46,84,85,88–93]. In our work, we tested the potential possibility of studying the state and phase transitions of a large and complex multiprotein, the photosystem II of higher plants (containing peptides, lipids, photosynthetic pigments, and electron transport cofactors). Photosystem II (PSII) is the pigment–protein complex located in the thylakoid membrane of chloroplasts and cyanobacteria. PSII catalyzes the transformation of light energy into electrochemical energy with separate charges [94]. A by-product of PSII's operation is molecular oxygen. A minimal molecular structure capable of water oxidizing is PSII core complexes. X-ray analysis of PS2 core complexes isolated from cyanobacteria showed that each PSII monomer consists of 20 protein subunits, 20 lipid molecules, as well as 35 chlorophyll and 12 carotenoid molecules [68,69,95,96]. Modernization of a fixed-length ultrasonic spectrometer, the acceleration of the speed, changes in the algorithm of determining the position of the resonance peak, and mixing the sample allowed performing measurements of heat-induced damage to PSII. In addition, such measurements became possible in the direction from low to high temperature without degassing of the measuring medium. Before the modernization, such measurements were not possible. Figure 6 indicates that heating of PSII core complexes led to changes in the state of the preparations, which led to the destruction of the complex. The first one started at 33.5 °C (Figure 6C), the second at 43.5 °C, the third at 56.5 °C, and the fourth at 66.7 °C (Figure 6B). To simplify the discussion, we call these transitions A (43.5 °C), B1 (56.5 °C), C (66.7 °C) (the nomenclature accepted in [26] was used), and A2 (33.5 °C) (the nomenclature accepted in [28] was used). A sharp increase in the Q-factor value at 33.5 °C indicated changes in the size or shape of the particles in the suspension. Indeed, using differential scanning calorimetry and gel electrophoresis separation, the dissociation of the proteins of the PSII water-oxidizing complex (without protein denaturation) was previously demonstrated [28]. Transition A observed in Figure 6B may be due to the degradation of the WOC [26,28,97], PSII core monomerization [98], and damage to the PSII acceptor side with denaturation of 43 kDa, 28 kDa, and 22 kDa proteins [27,28]. Moreover, lipid phase transitions may occur at this temperature, but in PSII core complexes, its contribution should be minimal [99]. Practically the full disappearance of transition A in pre-illuminated particles may be due to photodamage to the corresponding components of PSII [100,101]. As has been shown earlier, transition B1 reflects denaturation of the proteins of the PSII core (47 kDa, D1 and D2) [26,28,102]. Surprisingly, the amplitude of transition B1 in pre-illuminated samples was several times higher than in untreated PSII core complexes. Now, it is difficult to explain what this is connected with. It is likely that 10 min of illumination of the samples in an

environment containing oxygen caused significant damage to PSII's components. These damages can lead to a change in the properties of polypeptides due to their oxidation by reactive oxygen species [51,75,103–117], as well as the formation of cross-links between them [118–123]. On the one hand, the damages can reduce the thermal stability of the peptides, and on the other hand, cross-links can increase it. It is likely that during photoinhibition, additional covalent cross-links of the protein complex appear, leading to an increase in thermal stability. As a result of photoinhibition, the process of the destruction of the complex observed in the control samples at 44–52 °C may be shifted to the region of 57–60 °C and occurs in a much narrower temperature range. Note that the properties of untreated and pre-illuminated particles became similar after ≈60 °C. It is believed that transition C (identical in both types of preparations) reflects the degradation of minor light-harvesting complexes [20]. However, the PSII core complexes used in the present work did not contain these components. We assumed that transition C may reflect the denaturation of the PsbO protein of the water-oxidizing complex. Phase transitions of PsbO were observed at temperatures from 68 °C to 76 °C, depending on the presence of calcium and manganese ions and the pH of the medium [124].

Our data show that a fixed-length ultrasonic interferometry can be used not only to analyze simple compounds, but also complex ones, such as photosynthetic complexes isolated from the thylakoid membranes of higher plants, as well as the kinetics of heat- and photo-induced processes in them.

## 5. Conclusions

The present study showed the high sensitivity of the ultrasonic spectrometer we developed. The spectrometer proved itself in studies of both relatively simple compounds (for example, solutions of deuterium oxide and liposomes) and extremely complex multicomponent enzymes (for example, the core complexes of photosystem II of higher plants). Thus, the present study demonstrates that a fixed-length ultrasonic spectrometer can be applied in further investigations to determine the kinetics of heat- and photo-induced damage to biological objects.

**Author Contributions:** Conceptualization, D.V.Y., M.E.A. and S.V.G.; methodology, D.V.Y. and M.E.A.; software, M.E.A.; formal analysis, D.V.Y. and M.E.A.; investigation, D.V.Y., M.E.A., A.A.K., M.O.P. and D.A.S.; writing—original draft preparation, D.V.Y.; writing—review and editing, D.V.Y.; funding acquisition, D.V.Y. All authors have read and agreed to the published version of the manuscript.

**Funding:** The work was supported by the Russian Science Foundation Grant No. 22-24-01179, https://rscf.ru/project/22-24-01179/ (accessed on 29 September 2022).

**Institutional Review Board Statement:** Not applicable.

**Informed Consent Statement:** Not applicable.

**Data Availability Statement:** Not applicable.

**Conflicts of Interest:** The authors declare no conflict of interest.

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
