# Peer review of "Application of Fixed-Length Ultrasonic Interferometry to Determine the Kinetics of Light-/Heat-Induced Damage to Biological Membranes and Protein Complexes"

_inventions, doi:10.3390/inventions7040087_

Round 1

Reviewer 1 Report

The authors report of an alternative experimental approach to detect phase transitions and photoinduced damage of lipid membranes, which will be certainly of interest to the biophysical and biosensing community. The paper deserves to be published after some questions are addressed:

1) The authors should comment in the introduction that phospholipids, in particular, the phospholipid they use, can also show a ripple phase apart from the gel and the liquid disordered phase. They can cite P. Losada-Pérez et al., Phase transitions in lipid vesicles detected by a complementary set of methods: heat-transfer measurements, adiabatic scanning calorimetry and dissipation-mode quartz crystal microbalance, Physica Status Solidi A 211, 2014, 1377

2) The phase transition width of DPPC depends on the lamellarity of the lipid vesicles (melting of multilamellar vesicles is more cooperative) and the phase transition width can be less than 1 °C (see above-mentioned reference). The authors should clarify if their vesicles were multilamellar, unilamellar or both based on their extrusion pore diameter and number of extrusion passes.

3) The authors mention as well that mention that alcohols decrease the phase transition but do not provide any reference. They should provide a reference for this, i.e., S. Neupane et al., Real-time monitoring of interactions between solid-supported lipid vesicle layers and short- and medium- chain length alcohols: Ethanol and 1-pentanol, Biomimetics 4, 2019, 8

4) What is the concentration of lipids used in such a small volume? When citing techniques for measuring phospholipid phase transitions, the authors should include recent advances as well, such as quartz crystal microbalance with dissipation, which detects transitions accurately and with the need of a very small sample needed, see S. Neupane et al., Quartz crystal microbalance with dissipation monitoring: A versatile tool to monitor phase transitions in biomimetic membranes, Frontiers in Materials 5, 2018, 46

5) How can the authors determine excess enthalpy with ultrasonic measurements?

6) How do their results compare with commercial speed of sound measurements? The authors should compare their results to previously reported ultrasonic measurements, see for instance, S. Halstenberg et al., Cholesterol-induced variations in the volume and enthalpy fluctuations of lipid bilayers, Biophysical Journal 75, 1998, 264

7) Why are two peaks observed when palmitic acid is added? What does it imply at the level of phospholipid organization?

Author Response

Dear Reviewer 1,

We are very thankful for your comments.

The authors report of an alternative experimental approach to detect phase transitions and photoinduced damage of lipid membranes, which will be certainly of interest to the biophysical and biosensing community. The paper deserves to be published after some questions are addressed:

1) The authors should comment in the introduction that phospholipids, in particular, the phospholipid they use, can also show a ripple phase apart from the gel and the liquid disordered phase. They can cite P. Losada-Pérez et al., Phase transitions in lipid vesicles detected by a complementary set of methods: heat-transfer measurements, adiabatic scanning calorimetry and dissipation-mode quartz crystal microbalance, Physica Status Solidi A 211, 2014, 1377

2) The phase transition width of DPPC depends on the lamellarity of the lipid vesicles (melting of multilamellar vesicles is more cooperative) and the phase transition width can be less than 1 °C (see above-mentioned reference). The authors should clarify if their vesicles were multilamellar, unilamellar or both based on their extrusion pore diameter and number of extrusion passes.

3) The authors mention as well that mention that alcohols decrease the phase transition but do not provide any reference. They should provide a reference for this, i.e., S. Neupane et al., Real-time monitoring of interactions between solid-supported lipid vesicle layers and short- and medium- chain length alcohols: Ethanol and 1-pentanol, Biomimetics 4, 2019, 8

4) What is the concentration of lipids used in such a small volume? When citing techniques for measuring phospholipid phase transitions, the authors should include recent advances as well, such as quartz crystal microbalance with dissipation, which detects transitions accurately and with the need of a very small sample needed, see S. Neupane et al., Quartz crystal microbalance with dissipation monitoring: A versatile tool to monitor phase transitions in biomimetic membranes, Frontiers in Materials 5, 2018, 46

5) How can the authors determine excess enthalpy with ultrasonic measurements?

6) How do their results compare with commercial speed of sound measurements? The authors should compare their results to previously reported ultrasonic measurements, see for instance, S. Halstenberg et al., Cholesterol-induced variations in the volume and enthalpy fluctuations of lipid bilayers, Biophysical Journal 75, 1998, 264

7) Why are two peaks observed when palmitic acid is added? What does it imply at the level of phospholipid organization?

Below are the answers to your comments.

1) The authors should comment in the introduction that phospholipids, in particular, the phospholipid they use, can also show a ripple phase apart from the gel and the liquid disordered phase. They can cite P. Losada-Pérez et al., Phase transitions in lipid vesicles detected by a complementary set of methods: heat-transfer measurements, adiabatic scanning calorimetry and dissipation-mode quartz crystal microbalance, Physica Status Solidi A 211, 2014, 1377

We have made changes to new version of the manuscript in accordance with your comments.

2) The phase transition width of DPPC depends on the lamellarity of the lipid vesicles (melting of multilamellar vesicles is more cooperative) and the phase transition width can be less than 1 °C (see above-mentioned reference). The authors should clarify if their vesicles were multilamellar, unilamellar or both based on their extrusion pore diameter and number of extrusion passes.

The large unilamellar dimyristoylphosphatidylcholine vesicles were used in present work. The vesicles were prepared by extrusion fifteen times through a 200 nm membrane at room temperature. Prior to this, the dimyristoylphosphatidylcholine was hydrated for several hours, followed by five freezing/thawing cycles. It is known that such a procedure leads to the formation of unilamelar vesicles. [42; MacDonald, R.C.; MacDonald, R.I.; Menco, B.P.M.; Takeshita, K.; Subbarao, N.K.; Hu, L.-r. Small-volume extrusion apparatus for preparation of large, unilamellar vesicles. Biochimica et Biophysica Acta (BBA) - Biomembranes 1991, 1061, 297-303, doi:https://doi.org/10.1016/0005-2736(91)90295-J.; Preparing Large, Unilamellar Vesicles by Extrusion (LUVET). Available online: https://avantilipids.com/tech-support/liposome-preparation/luvet (accessed on 13 September 2022)]. We have made changes to new version of the manuscript in accordance with your comments.

3) The authors mention as well that mention that alcohols decrease the phase transition but do not provide any reference. They should provide a reference for this, i.e., S. Neupane et al., Real-time monitoring of interactions between solid-supported lipid vesicle layers and short- and medium- chain length alcohols: Ethanol and 1-pentanol, Biomimetics 4, 2019, 8

We have made changes to new version of the manuscript in accordance with your comments.

4) What is the concentration of lipids used in such a small volume? When citing techniques for measuring phospholipid phase transitions, the authors should include recent advances as well, such as quartz crystal microbalance with dissipation, which detects transitions accurately and with the need of a very small sample needed, see S. Neupane et al., Quartz crystal microbalance with dissipation monitoring: A versatile tool to monitor phase transitions in biomimetic membranes, Frontiers in Materials 5, 2018, 46

The concentration of lipids used our measurements was 1 mg/mL. We have made changes to new version of the manuscript in accordance with your comments.

5) How can the authors determine excess enthalpy with ultrasonic measurements?

We are aware of the fundamental possibility of estimating the added enthalpy using ultrasonic velocity measurements. However, for this, the experiments must be supplemented with measurements in a precision densitometer and a viscometer (or rheometer). For example, Vanathi, V.; Mullainathan, S.; Nithiyanatham, S.; Ramasamy, V.; Palaniappan, L. Ultrasonic velocity, density, viscosity for the ternary mixture of (benzene + chloroform + cyclohexane) at different temperatures. Heliyon 2019, 5, e02203, doi:https://doi.org/10.1016/j.heliyon.2019.e02203. However, this goes far beyond the limits of the task that we set ourselves in this work.

6) How do their results compare with commercial speed of sound measurements? The authors should compare their results to previously reported ultrasonic measurements, see for instance, S. Halstenberg et al., Cholesterol-induced variations in the volume and enthalpy fluctuations of lipid bilayers, Biophysical Journal 75, 1998, 264

Our data is consistent with the previous data, including those obtained using ultrasonic measurements. The point of change of the phase state of DMPC liposomes and phase transition width correspond to those obtained in [S. Halstenberg et al., Cholesterol-induced variations in the volume and enthalpy fluctuations of lipid bilayers, Biophysical Journal 75, 1998, 264]. We have made changes to new version of the manuscript in accordance with your comments.

7) Why are two peaks observed when palmitic acid is added? What does it imply at the level of phospholipid organization?

The formation of the second peak may be due to the formation of a population of a more stable domain in the presence of high concentrations of PA. For example, in [Saitta, F.; Signorelli, M.; Fessas, D. Dissecting the effects of free fatty acids on the thermodynamic stability of complex model membranes mimicking insulin secretory granules. Colloids and Surfaces B: Biointerfaces 2019, 176, 167-175, doi:https://doi.org/10.1016/j.colsurfb.2018.12.066] using the micro-DSC technique, the effect of PA on the thermal stability of phospholipid vesicles was shown. PA has been shown to have a strong stabilizing effect on membranes. When the PA concentration reaches 25%, the micro-DSC thermogram shows the formation of a second peak, probably associated with the enrichment of the population of a more stable domain.

Thank you again for your comments, which helped us a lot to improve the manuscript.

Reviewer 2 Report

The manuscript “Application of fixed length ultrasonic interferometry to determine the kinetics of the light/heat-induced damage to the biological membranes and protein complexes” presented by Yanykin et al describes the kinetics of heat- and photo-induced damage to biological membranes and protein complexes determination using a custom-made fixed-length ultrasonic spectrometer.

Overall, the presented study is very interesting as authors indicate the availability of a low cost fixed-length ultrasonic spectrometer that will allow the partial characterization of the of lipid bilayer membrane in the absence and presence of protein complexes. This might be an additional tool for the measurement of membrane mechanics next to more traditional compression of lipid monolayers, differential scanning calorimetry or electron paramagnetic resonance. However, as authors want to present a proof-of-concept and draw a general conclusion for the application of their new “tool”, they need to present a larger dataset, that on one hand validates their results and conclusions in comparison with more traditional methods and on the other hand as authors correlate their measurements with lipid phase transitions of gel-to-sol states, they should be able to distinguish different types of liposomes composed of one type of lipid , when possible (specific phospholipids as PS, PG; PE , PI, Cl; ceramides; shingolipids,…)  or liposomes composed of several lipids as mimics of natural membranes.

Next, though the authors show the characterization of a proteoliposome-reconstituted integral protein complex (Photosystem II), they need to compare their data with e.g. blue native gel electrophoresis or fluorescence spectroscopy to correlate their findings with the unfolding of protein domains or complete protein denaturalization. A literature search will provide authors with experimental details. In addition, I think it would also be interesting to use commercial pore forming peptides like alamethicin or melittin next to complex integral membrane protein and record their oligomeric state during insertion, pore formation and membrane collapse on lipid bilayer membrane vesicles. The presented technique should be able to visualize transition states of this peptide-lipid and peptide-peptide interactions.    

A complete data set would strengthen the potential of the tool developed by authors and help it to be adopted by the scientific community

Author Response

Dear Reviewer 2,

We are very thankful for your comments.

The manuscript “Application of fixed length ultrasonic interferometry to determine the kinetics of the light/heat-induced damage to the biological membranes and protein complexes” presented by Yanykin et al describes the kinetics of heat- and photo-induced damage to biological membranes and protein complexes determination using a custom-made fixed-length ultrasonic spectrometer.

Overall, the presented study is very interesting as authors indicate the availability of a low cost fixed-length ultrasonic spectrometer that will allow the partial characterization of the of lipid bilayer membrane in the absence and presence of protein complexes. This might be an additional tool for the measurement of membrane mechanics next to more traditional compression of lipid monolayers, differential scanning calorimetry or electron paramagnetic resonance. However, as authors want to present a proof-of-concept and draw a general conclusion for the application of their new “tool”, they need to present a larger dataset, that on one hand validates their results and conclusions in comparison with more traditional methods and on the other hand as authors correlate their measurements with lipid phase transitions of gel-to-sol states, they should be able to distinguish different types of liposomes composed of one type of lipid , when possible (specific phospholipids as PS, PG; PE , PI, Cl; ceramides; shingolipids,…)  or liposomes composed of several lipids as mimics of natural membranes.

Next, though the authors show the characterization of a proteoliposome-reconstituted integral protein complex (Photosystem II), they need to compare their data with e.g. blue native gel electrophoresis or fluorescence spectroscopy to correlate their findings with the unfolding of protein domains or complete protein denaturalization. A literature search will provide authors with experimental details. In addition, I think it would also be interesting to use commercial pore forming peptides like alamethicin or melittin next to complex integral membrane protein and record their oligomeric state during insertion, pore formation and membrane collapse on lipid bilayer membrane vesicles. The presented technique should be able to visualize transition states of this peptide-lipid and peptide-peptide interactions.

A complete data set would strengthen the potential of the tool developed by authors and help it to be adopted by the scientific community

Below are the answers to your comments.

  1.  However, as authors want to present a proof-of-concept and draw a general conclusion for the application of their new “tool”, they need to present a larger dataset, that on one hand validates their results and conclusions in comparison with more traditional methods and on the other hand as authors correlate their measurements with lipid phase transitions of gel-to-sol states, they should be able to distinguish different types of liposomes composed of one type of lipid , when possible (specific phospholipids as PS, PG; PE , PI, Cl; ceramides; shingolipids,…) or liposomes composed of several lipids as mimics of natural membranes.

Similar studies were done in other works, both before the modernization of the installation, and by other authors. It has been shown that the ultrasound method is able to distinguish both different types of liposomes composed of one type of lipid and specific phospholipids as well as liposomes composed of several lipids [Astashev, M.; Belosludtsev, K.; Kharakoz, D. Method for digital measurement of phase-frequency characteristics for a fixed-length ultrasonic spectrometer. Acoustical Physics 2014, 60, 335-341; Halstenberg, S.; Heimburg, T.; Hianik, T.; Kaatze, U.; Krivanek, R. Cholesterol-induced variations in the volume and enthalpy fluctuations of lipid bilayers. Biophysical journal 1998, 75, 264-271; Taylor, T.M.; Davidson, P.M.; Bruce, B.D.; Weiss, J. Ultrasonic spectroscopy and differential scanning calorimetry of liposomal-encapsulated nisin. Journal of agricultural and food chemistry 2005, 53, 8722-8728].

  1. Next, though the authors show the characterization of a proteoliposome-reconstituted integral protein complex (Photosystem II), they need to compare their data with e.g. blue native gel electrophoresis or fluorescence spectroscopy to correlate their findings with the unfolding of protein domains or complete protein denaturalization. A literature search will provide authors with experimental details.

It should be noted that in our work we used the PSII Core complexes in an aqueous solution containing a small amount of a “gentle” detergent. The complexes were not incorporated into proteoliposomes. In the discussion, we interpreted the data obtained in this work based on the analysis of previously published data obtained using differential scanning calorimetry, thermal gel analysis, analysis of loss of PSII activity, heat disassmbly of the PSII complex , electron microscopy, fluorescence spectroscopy, the near-UV circular dichroism etc. [Thompson, L.K.; Blaylock, R.; Sturtevant, J.M.; Brudvig, G.W. Molecular basis of the heat denaturation of photosystem II. Biochemistry 1989, 28, 6686-6695; Shutova, T.; Nikitina, J.; Deikus, G.; Andersson, B.; Klimov, V.; Samuelsson, G. Structural dynamics of the manganese-stabilizing protein effect of pH, calcium, and manganese. Biochemistry 2005, 44, 15182-15192; Thompson, L.K.; Sturtevant, J.M.; Brudvig, G.W. Differential scanning calorimetric studies of photosystem II: evidence for a structural role for cytochrome b559 in the oxygen-evolving complex. Biochemistry 1986, 25, 6161-6169; Shutilova, N.; Semenova, G.; Klimov, V.; Shnyrov, V. Temperature-induced functional and structural transformations of the photosystem II oxygen-evolving complex in spinach subchloroplast preparations. Biochem Mol Biol Int 1995, 35, 1233-1243 and others].

  1.  In addition, I think it would also be interesting to use commercial pore forming peptides like alamethicin or melittin next to complex integral membrane protein and record their oligomeric state during insertion, pore formation and membrane collapse on lipid bilayer membrane vesicles. The presented technique should be able to visualize transition states of this peptide-lipid and peptide-peptide interactions.

For studies using commercial pore forming peptides like alamethicin or melittin, PSII Core complexes must be incorporated into a liposome (in our work we used the PSII Core complexes in an aqueous solution). By creating an electrically closed system, we will be able to conduct research using pore forming agents. Your question is very interesting and will be of great help to our future research. We will conduct relevant research in the next research, and hope to obtain interesting results.

Thank you again for your comments, which will be of great help to us in our subsequent experiments.

Round 2

Reviewer 2 Report

Authors superficially addressed the raised concern. Tough they state that raised concerns were already answered in other publications, in my opinion it is important that they "again" demonstrate that it holds true for their system. 

Statements throughout the text as "probably due..." are speculative and should be avoided by validation through additional methods. Especially when presenting a useful tool.

In my opinion, authors demonstrate a lack of rigor and respect to  the scientific community.  

Though I like the ms and think that the finding are important and should be published, I will not recommend it for publication.  The editor should seek another reviewer for final evaluation.